# APOA1 Is a Novel Marker for Preeclampsia

**DOI:** 10.3390/ijms242216363

**Published:** 2023-11-15

**Authors:** Zhenzhen Liu, Jiangnan Pei, Xiaoyue Zhang, Chengjie Wang, Yao Tang, Haiyan Liu, Yi Yu, Shouling Luo, Weirong Gu

**Affiliations:** 1Department of Obstetrics and Gynecology, Obstetrics and Gynecology Hospital of Fudan University, Shanghai 200011, China; 21211250010@m.fudan.edu.cn (Z.L.); 19111250017@fudan.edu.cn (J.P.); 22211250022@m.fudan.edu.cn (X.Z.); wangcj18@fudan.edu.cn (C.W.); tangtang061114158@163.com (Y.T.); haiyanliu11@fudan.edu.cn (H.L.); 12301010077@fudan.edu.cn (Y.Y.); 15211250009@fudan.edu.cn (S.L.); 2Shanghai Key Laboratory of Female Reproductive Endocrine Related Diseases, Shanghai 200011, China

**Keywords:** preeclampsia, apolipoprotein A1, PPARγ, trophoblast

## Abstract

Preeclampsia (PE) is one of the pregnancy complications, leading to major maternal and fetal morbidity and mortality; however, the underlying mechanisms of PE still remain unclear. We aimed to explore the role of apolipoprotein A1 (APOA1) in the pathophysiology of PE. The expression of APOA1 was elevated in both plasma and placental tissues, as detected by Western blotting, immunohistochemistry, and a qRT-PCR assay. Importantly, we detected the concentration of APOA1 using the ELISA assay in normal control women (*n* = 30) and women with preeclampsia (*n* = 29) from a prospective cohort study. The concentration of APOA1 was not significantly altered in plasma during early and mid-term gestation of the PE patients compared to the NP patients; however, it was elevated during late gestation. Additionally, the concentration of APOA1 was positively associated with systolic blood pressure during late gestation. The proliferation and invasion of trophoblast were all increased in HTR8/SVneo cells transfected with *APOA1* siRNA and decreased in HTR8/SVneo cells treated with the recombinant human APOA1 protein (rhAPOA1). Additionally, we used public datasets to investigate the downstream genes of APOA1 and qRT-PCR for validation. Furthermore, we explored the transcriptional activity of peroxisome proliferator-activated receptor gamma (PPARγ) in APOA1 by using a luciferase assay, which showed that the *APOA1* promoter was activated by PPARγ. Additionally, the inhibitory effect of rhAPOA1 on the ability of trophoblast invasion and proliferation can be rescued by the PPARγ inhibitor. Our findings suggest the crucial role of APOA1 in PE, which might provide a new strategy for the prevention and treatment of PE.

## 1. Introduction

PE is a severe complication of pregnancy characterized by new-onset hypertension and proteinuria [1]. Inadequate trophoblast invasion and incomplete spiral artery remodeling are the key pathological features of the disease [2]. However, the underlying pathogenesis of PE remains largely unknown.

The role of apolipoproteins (apo) in the pathophysiology of PE has recently emerged [3,4,5,6,7,8]. Apolipoproteins constitute structural or regulatory components of lipoproteins, including HDL, VLDL, and LDL, as well as receptor ligands and cofactors for lipid-metabolizing enzymes [9]. Numerous biomarkers have been found to be associated with PE, including different apolipoproteins [3,4,5]. Evidence suggests that levels of HDL-associated apolipoprotein C-II in mothers are significantly increased in women with PE [6]. Furthermore, *APOC3* transgenic mice with abnormal lipid metabolism exhibit gestational hypertension [7]. Apolipoproteins APOA2, APOC1, APOC3, and APOE levels are decreased in women with GDM and may play a role in inflammation [8]. Thus, apolipoproteins play a pivotal role in the transport and metabolism of plasma lipids and may be involved in the pathophysiology of PE.

Apolipoprotein A1 (APOA1) is a major apoprotein constituent of high-density lipoproteins that plays important roles in tumor invasion and metastasis [10,11]. It has been reported that increased *lncRNA APOA1-AS* was found in the placental tissues of early onset severe PE, and downregulation of APOA1-AS protected against TNF-α-induced inhibition of trophoblast integration into endothelial networks, thus exerting protective effects in PE rats [12]. Recent findings have revealed the crucial roles of APOA1 in inflammation, tumor growth, angiogenesis, invasion, and metastasis [13,14,15]. For example, *APOA1* overexpression attenuates cell migration and proliferation and elevates apoptosis in SW1353 and HOS cells [16]. Evidence has revealed that low APOA1 expression is critical for establishing pregnancy, and elevated APOA1 expression in chorionic villi correlates with early miscarriage [17], which is also characterized by impaired trophoblast invasion. However, the exact role of APOA1 in PE remains to be elucidated.

In this study, we found that the expression of APOA1 was elevated in both the plasma and placental tissues of patients with preeclampsia. Importantly, the concentration of APOA1 was not significantly altered in plasma during early and mid-term gestation in the PE patients compared to the NP patients, whereas it was elevated during late gestation. Notably, the concentration of APOA1 was positively associated with systolic blood pressure during late gestation. Additionally, we found that the transcriptional activity of *APOA1* was regulated by PPARγ. Furthermore, APOA1 can inhibit trophoblast proliferation and invasion, which can be rescued by PPARγ. This study provides evidence for the crucial role of APOA1 in PE, which may provide a new strategy for the prevention and treatment of PE.

## 2. Results

### 2.1. APOA1 Was Significantly Elevated in the Placental and Plasma Samples from Women with PE

To evaluate the expression level of APOA1 in placental and plasma tissues from women with PE and NP, we performed immunohistochemistry, Western blotting, qRT-PCR, and ELISA assays (Figure 1A–E). The expression of APOA1 was significantly elevated in the placentas of women with PE compared to women with NP, both at the RNA and protein levels (Figure 1A,B,D). Next, the levels of APOA1 in plasma samples from NP and PE women were detected by Western blotting and ELISA. The results indicated that the expression of APOA1 in plasma samples was increased in women with PE detected by Western blotting (Figure 1C). Importantly, the expression of APOA1 in plasma tissues from women with PE was also significantly increased during late gestation, as detected by ELISA (Figure 1E). These results indicate that the expression of APOA1 was upregulated in the placental and plasma samples from women with PE.

### 2.2. APOA1 Cannot Predict the Risk of Preeclampsia

Because of the significant difference in APOA1 levels in placental and plasma tissues of PE and NP women, we aimed to explore the role of APOA1 as a biomarker for PE. We obtained data from 29 women with PE and 30 with NP from a prospective cohort. Subsequently, we detected the protein levels of APOA1 in plasma tissues during early, mid-term, and late gestation in 59 volunteers using ELISA. The results showed that there was no significant difference in APOA1 levels in the plasma of patients at early and mid-term gestation. In contrast, the level of APOA1 in the plasma of women with PE during late gestation was significantly higher than that in women with NP (Figure 1E). Additionally, we detected the concentration of PLGF and sFlt1 in the plasma tissues of 59 volunteers and found that the ratio of sFlt1/PLGF was dramatically increased in early, mid-term, and late gestation of women with PE (Figure 1F).

We found that the expression level of APOA1 in plasma from early and mid-term gestation was not significantly associated with systolic blood pressure in the patients (Figure 1G,H). In contrast, the expression level of APOA1 in the plasma from late gestation was positively associated with the systolic blood pressure of the patients (Figure 1I). Additionally, we found that the ratio of sFlt1/PLGF [0.691 (95%CI: 0.551, 0.83), *p* = 0.012, during early gestation; 0.926 (95%CI: 0.860, 0.992), *p* < 0.0001, during mid-term gestation] exhibited better efficiency for predicting the risk of preeclampsia than the level of APOA1 [0.521, (95%CI: 0.351, 0.69), *p* = 0.785, during early gestation; 0.555, (95%CI: 0.392, 0.717), *p* = 0.474, during mid-term gestation]. However, the prediction efficiency was not statistically altered by the combination of the sFlt1/PLGF and the level of APOA1, both in early and mid-term gestation (Figure 1J,K).

### 2.3. APOA1 Can Inhibit the Proliferation and Invasion of Trophoblast Cells

To validate the function of APOA1, trophoblast cell lines were used for further experiments. HTR8/SVneo, Bewo, and Jeg3 are three common trophoblast cell lines. qRT-PCR assays revealed that the expression of *APOA1* was highest in Jeg3, lowest in Bewo, and intermediate in HTR8/SVneo cells (Figure 2A). In addition, HTR8/SVneo cells are immortal extravillous trophoblast cells derived from normal first-trimester placentas [18]; therefore, HTR8/SVneo cells were used in subsequent experiments.

To further verify the mechanisms of APOA1 in preeclampsia, we knocked down *APOA1* in HTR8/SVneo cells using three different siRNAs. The expression of *APOA1* was significantly downregulated after transfection with the three siRNAs, with *APOA1*-si-2 having the most significant effect (Figure 2B). Therefore, we selected *APOA1*-si-2 for further experiments. Trophoblast invasion and proliferation were significantly increased after *APOA1* knockdown, while they were decreased in HTR8/SVneo cells treated with rhAPOA1 protein (Figure 2C,D).

### 2.4. APOA1 Can Be Transcriptionally Regulated by PPARγ

To explore the interaction between APOA1 and PPARγ, we treated HTR8/SVneo cells with the PPARγ agonist rosiglitazone and PPARγ inhibitor T0070907. *PPARγ* expression was downregulated in HTR8/SVneo cells treated with T0070907, whereas it was upregulated in HTR8/SVneo cells treated with rosiglitazone (Figure 3A). Similarly, the expression of *APOA1* decreased in HTR8/SVneo cells treated with T0070907 and increased in HTR8/SVneo cells treated with rosiglitazone (Figure 3B). Furthermore, we found that the expression of *APOA1* and *PPARγ* was downregulated in *PPARγ*-KD cells transfected with siRNA (Figure 3D,E).

To further examine the transcriptional activity of PPARγ in the *APOA1* promoter, we performed a luciferase assay using a dual-luciferase reporter system in HTR8/SVneo and ERK 293T cells (Figure 3F,G). The results revealed that the *APOA1* promoter was active after PPARγ overexpression, whereas it was inhibited after the mutation of the APOA1 promoter. 

In addition, the inhibition of proliferation and invasive ability of trophoblast cells by rhAPOA1 protein was rescued by the PPARγ inhibitor T0070907 (Figure 2C,D). These results indicate that APOA1 is transcriptionally regulated by PPARγ.

### 2.5. APOA1 Functions by Regulating Several Downstream Targets

Public datasets were used for further analysis to explore the downstream targets of APOA1. These datasets were mRNA sequences obtained from *APOA1* over-expressed 4T cells and wild-type 4T cells. We identified 9451 genes from *APOA1* overexpression and wild-type 4T cells. Of these, 70 DEGs were identified and visualized using a heatmap (Appendix A), among which 28 were downregulated and 42 were upregulated. Additionally, a PPI network was constructed (Appendix A). KEGG enrichment analysis showed that these 70 DEGs were mainly associated with ECM-receptor interactions (Appendix A). Furthermore, 40 GO biological processes were identified, and the top 10 terms are shown in Appendix A. These terms included cell adhesion, regulation of the apoptotic process, regulation of cell proliferation, and cell migration.

Next, we selected several significant DEGs for validation by qRT-PCR in placentas from patients with PE and NP, including *MMP3* (matrix metallopeptidase 3), *ITGB2* (integrin beta2), *COL6A1* (Collagen type VI alpha 1), and *LRP1* (low-density lipoprotein receptor-related protein 1). These genes were all increased in *APOA1* over-expressed 4T cells (Figure 4A–D). Among these, *MMP3* was most significantly elevated in APOA1 overexpressed groups, with an approximately 3.6-fold increase (Figure 4A). In addition, these genes were elevated in the placentas of preeclampsia patients (Figure 4E–H), including *LRP1* (Figure 4E), *COL6A1* (Figure 4F), *ITGB2* (Figure 4G), and *MMP3* (Figure 4H). Interestingly, *MMP3* still has the largest increase, with about a 2.6-fold increase. Therefore, the effect of MMP3 was the most significant. Furthermore, we investigated the function of MMP3 by performing CCK-8 and Transwell assays. The results showed that the proliferation and invasion of trophoblast cells were elevated after treatment with MMP3 inhibitor 1 but could be rescued by adding rhAPOA1 protein (Figure 2C,D). The expression of *MMP3* decreased in HTR8/SVneo cells treated with T0070907 and increased in HTR8/SVneo cells treated with rosiglitazone (Figure 3C). These results indicate that APOA1 might inhibit trophoblast proliferation and invasion by regulating these genes, particularly *MMP3*.

In addition, we investigated the regulatory role of APOA1 in these four genes. First, we treated HTR8/SVneo cells with rhAPOA1 and the PPARγ inhibitor T0070907 and then detected the expression of *LRP1*, *COL6A1*, *ITGB2*, *MMP3*, and *APOA1*. The results showed that the expression of *APOA1* increased (Figure 5E) in HTR8/SVneo cells treated with rhAPOA1, suggesting the effect of rhAPOA1 on the expression of *APOA1*. Furthermore, high expression of *APOA1* in HTR8/SVneo cells treated with rhAPOA1 was attenuated by T0070907 (Figure 5E). The expression of *COL6A1* (Figure 5B), and *MMP3* (Figure 5D) increased slightly in HTR8/SVneo cells treated with rhAPOA1, which can be rescued by T0070907 supplementation. However, the expression of *LRP1* increased slightly in HTR8/SVneo cells treated with rhAPOA1, which cannot be rescued by T0070907 (Figure 5A). The expression of *ITGB2* was not significantly changed in *APOA1*-overexpressed HTR8/SVneo cells, while that was inhibited in HTR8/SVneo cells treated with T0070907 (Figure 5C). Subsequently, we treated HTR8/SVneo cells with *APOA1* siRNA and PPARγ agonist rosiglitazone and then detected the expression of *LRP1*, *COL6A1*, *ITGB2*, *MMP3*, and *APOA1*. The results showed that the expression of *APOA1* was decreased (Figure 5J) in HTR8/SVneo cells treated with *APOA1* siRNA, which can be rescued by rosiglitazone, suggesting an inhibitory effect of APOA1 siRNA on the expression of APOA1. The expression of *LRP1* (Figure 5F), *COL6A1* (Figure 5G), and *MMP3* (Figure 5I) decreased significantly, which can be elevated by supplementation with rosiglitazone. However, the low expression of *ITGB2* in *APOA1*-knockdown HTR8/SVneo cells was not rescued by rosiglitazone (Figure 5H).

## 3. Discussion

In the present study, we investigated the role of APOA1 in PE pathophysiology. We found that the expression of APOA1 was significantly elevated in the plasma of women with preeclampsia during late gestation but not during early and mid-term gestation. Importantly, we found that the ratio of sFlt1/PLGF was more efficient in predicting the risk of preeclampsia than the level of APOA1. However, the prediction efficiency was not significantly altered by the combination of sFlt1/PLGF and APOA1 levels, both in early and mid-term gestation. These results suggested that APOA1 may not function in predicting the risk of PE. Additionally, the concentration of APOA1 was positively associated with systolic blood pressure during late gestation. Overexpression of APOA1 exerted inhibitory effects on trophoblast cell proliferation and invasion, which could be rescued by a PPARγ inhibitor. Mechanistically, the transcriptional activity of APOA1 is mediated by PPARγ.

Evidence suggests that abnormal lipid metabolism is involved in preeclampsia [19]. Additionally, research on serum lipid biomarkers for predicting PE has recently emerged. For example, it has been reported that developed panels of serum lipidomic biomarkers can identify most women at risk for preeclampsia in a given pregnancy at 12–14 weeks of gestation [20]. Additionally, numerous biomarkers have been evaluated for their association with PE, including various apolipoproteins [3,4,5,21]. In this study, we found that the plasma concentration of APOA1 in early and mid-term gestation was not statistically different, whereas it was significantly increased in late gestation. Additionally, the sFlt1/PLGF ratio was markedly elevated in the plasma of women with preeclampsia during early, mid-term, and late gestation. Notably, the AUC of the ratio of sFlt1/PLGF during mid-term gestation for predicting preeclampsia was 0.926 (95%CI: 0.862–0.992), while it was not significantly elevated by adding the concentration of APOA1. These results suggest that the alteration in the expression of APOA1 might be caused by the pathophysiology of PE. Thus, it is not realistic to use APOA1 as a predictor of preeclampsia onset; however, APOA1 may be useful as a judgment of the severity of the condition, which in turn may help in the treatment of the disease.

APOA1 overexpression was associated with adverse pregnancy outcomes. In this study, we found that APOA1 levels were elevated in both the plasma and placental tissues of patients with preeclampsia. It has been reported that the expression of APOA1 was high in the blood and maternal-fetal interface during early miscarriages [17]. Additionally, evidence has revealed that apolipoprotein A-1 antisense RNA was increased in the placental tissues of early onset severe PE, and downregulation of APOA1-AS protected against TNF-α-induced inhibition of trophoblast integration into endothelial networks, thus exerting protective effects against PE in rats [12]. This evidence indicates that the upregulation of APOA1 might indicate adverse pregnancy outcomes.

In addition, APOA1 inhibited cell proliferation and invasion. We found that the proliferation and invasion of trophoblast cells can be inhibited by treatment with rhAPOA1 and promoted by treatment with APOA1 siRNA. Research has also revealed that APOA1 regulates cellular growth, invasion, and apoptosis [16,22]. For example, a study demonstrated that cellular proliferation and invasion were inhibited in APOA1-overexpressed SW1353 and HOS cells, which is related to osteosarcoma progression [16]. Interestingly, we found that APOA1 functioned by regulating MMP3, COL6A1, ITGB2, and LRP1. Research has shown that levels of MMP3 in maternal plasma are significantly higher in preeclamptic patients with early onset disease [23]. These results suggest that APOA1 plays a role in cellular functions.

APOA1 may be regulated by PPARγ, which is involved in lipid metabolism. In this study, the data showed that the APOA1 promoter can be activated by PPARγ, and PPARγ agonists can promote the expression of APOA1. Previous reports have shown that the transcription factor PPARγ is essential for placental development and alterations in its expression and/or activity are associated with human placental pathologies such as preeclampsia [24]. Furthermore, PPARα activators, such as normolipidemic fibric acids, decrease triglyceride concentrations by increasing lipoprotein lipase expression and decreasing APOC3 concentration. They then increase HDL cholesterol levels by increasing the expression of APOA1 and APOA2 [25]. This evidence indicates an interaction between APOA1 and PPARγ.

However, our study has some limitations. We used RNA sequencing data from APOA1-overexpressed 4T cells from GEO datasets. We validated only five DEGs using qRT-PCR. In conclusion, our study provides a new approach to exploring the mechanisms underlying preeclampsia. In particular, APOA1 may be useful for judging the severity of the condition, which in turn may help in the treatment of the disease. This study provides evidence for the crucial role of APOA1 in PE, which may provide a new strategy for the prevention and treatment of PE.

## 4. Materials and Methods

### 4.1. Human Samples

#### 4.1.1. Subjects of the Study

The subjects of this study were obtained from a prospective observational cohort under the National Key Research and Development Program “Research on Reproductive Health and Prevention and Control of Major Birth Defects”, Subtopic III “Intervention of Preeclampsia”. The cohort recruited singleton pregnant women of advanced age (≥35 years) who were enrolled between September 2017 and December 2021 at the Obstetrics and Gynecology Hospital of Fudan University (Shanghai, China). The study was approved by the Ethics Committee of the Obstetrics and Gynecology Hospital Affiliated to Fudan University (Shanghai, China) (2017-16, 20170424). All the volunteers provided informed consent. We randomly selected 30 women with normal pregnancies (NP) and 29 with PE. Women with PE were obtained according to the criteria of the American College of Obstetricians and Gynecologists [26]. Table 1 presents the clinical characteristics of the 59 participants.

#### 4.1.2. Inclusion and Exclusion Criteria

Eligible participants were women who were recruited at ≤14 gestational weeks and were singleton pregnancies. We excluded women with a history of hypertension, nephritis, cardiac disease, multiple pregnancies, physical disability, current substance abuse, or any other conditions that may be intolerant to pregnancy.

#### 4.1.3. Samples Collection and Testing

Human plasma samples were collected from 29 PE and 30 NP patients during their first, second, and third trimesters for ELISA analysis. The expression of APOA1, placental growth factor (PLGF), and soluble fms-like tyrosine kinase-1 (sFlt1) in the plasma samples was detected using ELISA Kits (U96-2556E; U96-1590E; U96-1259E, YOBIBIO, Shanghai, China). In addition, we categorized the entire pregnancy into three stages: early gestation (first trimester), up to (but not including) 14 weeks; mid-term gestation (second trimester), 14–27^+6^ weeks; and late gestation (third trimester), 28 weeks and beyond.

Several placental samples (0.5 cm × 0.5 cm) were obtained immediately after delivery. Tissue was collected from the area near the umbilical cord (1 cm from the cord insert), from both the maternal and fetal sides of the placenta. After removal of the maternal blood cells by washing the tissue in sterile phosphate-buffered saline (PBS), the tissue was partially kept in MACS Tissue Storage Solution (Miltenyi Biotec, Bergisch-Gladbach, Germany) for qRT-PCR assay, partially fixed in 4% paraformaldehyde overnight, embedded in paraffin, and cut into 5 µm sections for immunohistochemistry tests.

### 4.2. Expression Profile Datasets Selection and Analysis

The datasets were obtained from the GEO DataSets portal, publicly available at the National Center for Biotechnology Information (NCBI) [27]. To identify the association between PPARγ and APOA1, we analyzed GSE206927 datasets, which included mRNA profiles of human aortic valvular endothelial cells transfected with siRNA targeting PPARγ (PPARγ-KD) or negative control siRNA (NC). To identify the genes downstream of APOA1, we analyzed the GSE202427 dataset, which is RNA sequencing data from APOA1 overexpressing 4T1 (APOA1_1, APOA1_2) and wild-type 4T1 (control_1, control_2) cells. Appendix A presents a list of differentially expressed genes for both data.

The DESeq2 (version 1.42.0) package in the R software (version 4.1.1) was used to analyze and identify differentially expressed genes (DEGs) with a threshold of |log2 (fold change) [FC]| ≥ 1.0, *p*-value < 0.05, after normalization and log2 transformation of the raw data. Heatmaps of the DEGs were plotted using the ggplot2 (version 3.4.4) package in the R software. The ClusterProfiler (version 4.10.0) package was used to perform biological functional enrichment analysis of DEGs using Gene Ontology (GO) and KEGG.

### 4.3. Real-Time Quantitative Polymerase Chain Reaction (qRT-PCR)

Total RNA was extracted from placental tissues and cells using an EZ-press RNA Purification Kit (B0004D; EZBioscience, Roseville, CA, USA). Next, the extracted RNA from all samples was diluted to 100 ng/μL for reverse transcription using Hifair^®^ III 1st Strand cDNA Synthesis SuperMix (11119ES, YEASEN, Shanghai, China). Real-time PCR was performed in triplicate using a QuantStudio™ 6 Flex Real-Time PCR System (Thermo Fisher Scientific, Waltham, MA, USA). The reaction mixtures (20 μL) contained 10 ng cDNA template, 200 nM reverse and forward primers, and 10 μL 2 × SYBR Green PCR Mix (11202ES03, YEASEN). The temperature profile included enzyme activation at 95 °C for 2 min, followed by 40 cycles of 10 s at 95 °C and 30 s at 60 °C, and a melting curve that included denaturation at 95 °C for 10 s, annealing at 60 °C for 1 min, high-resolution melting at 95 °C for 15 s, and annealing at 60 °C for 15 s. Glyceraldehyde 3-phosphate dehydrogenase (*GAPDH*) mRNA was used as a reference gene. The primers used for qRT-PCR are listed in Table 2. Relative mRNA expression was analyzed using the 2^−ΔΔCt^ method.

### 4.4. Cell Experiment

#### 4.4.1. Cell Culture

Trophoblastic HTR8/SVneo cells (CRL-3271, ATCC, Manassas, VA, USA) were cultured in DMEM/F12 medium with 10% fetal bovine serum (FBS) (Gibco, Pittsburgh, PA, USA) under standard culture conditions (37 °C and 5% CO_2_ in a humidified atmosphere).

Human embryonic kidney (HEK) 293T cells (CRL-3216, ATCC) were cultured in Dulbecco’s modified Eagle’s medium (DMEM) (SH30249.01, Hyclone, Logan, UT, USA) supplemented with 10% FBS and maintained under standard culture conditions.

#### 4.4.2. Small Interfering RNA (siRNA) Transfection

HTR8/SVneo cells were transfected with *APOA1* siRNA (GeneChem, Shanghai, China) using Lipofectamine 3000 (L3000075, Thermo Fisher Scientific) according to the manufacturer’s instructions. The siRNA sequences used were 5′-GUACGUGGAUGUGCUCAAA-3′ for *APOA1*-si-1, 5′-GAAGCUGCACGAGCUGCAA-3′ for *APOA1*-si-2, and 5′-GCUCUCGAGGAGUACACUA-3′ for *APOA1*-si-3.

#### 4.4.3. Cell Treatment

HTR8/SVneo cells were treated with emapticap pegol MMP3 inhibitor 1 (1 µM, HY-114418, MedChemExpress, Monmouth Junction, NJ, USA), T0070907 (10 µM, S2871, Selleck Chemicals, Houston, TX, USA), rosiglitazone (10 µM, S2556, Selleck Chemicals), rhAPOA1 (HY-P7525, MedChemExpress), and 0.1% DMSO for 24 h. The optimal concentrations of T0070907 and rosiglitazone were determined as previously described [28], whereas those of MMP3 inhibitor 1 and rhAPOA1 were determined using the cell counting kit-8 (CCK-8) assay.

#### 4.4.4. Transwell Assay

Matrigel (356234, BD Biosciences, San Diego, CA, USA) was diluted at 1:8 and added to the upper chamber of the 24-well plate with 8.0 µm transparent PET membrane (353097, Corning, Palo Alto, CA, USA) overnight at 4 °C. Next, 200 µL (1 × 10^5^ HTR8/SVneo cells) of DMEM/F12 suspension without FBS was added to the upper chamber, and 700 µL of DMEM/F12 containing 10% FBS was added to the lower chamber. The cells were cultured at 37 °C in a 5% CO_2_ incubator for 48 h. The 24-well plate was removed, and the upper chamber medium and non-penetrating cells were gently wiped off with a cotton swab, washed three times with PBS, fixed with methanol for 5 min, and stained with crystal violet for 20 min. Subsequently, random photographs were acquired under an inverted microscope (×200), and five visual fields were counted in each chamber.

#### 4.4.5. Cell Proliferation Assay

Cell counting kit-8 (40203ES, YEASEN) was used to test the cell proliferation ability according to the manufacturer’s instructions. HTR8/SVneo cells (96-well plates, 1 × 10^4^ cells/well) were treated with vehicle, rhAPOA1, *APOA1*-si-2, MMP3 inhibitor 1, rhAPOA1 + MMP3 inhibitor 1, or rhAPOA1 + T0070907 for 48 h. A 96-well plate was then cultured with 10 μL/well CCK-8 solution for 1.5 h. The absorbance was measured at 450 nm using a microplate reader (Bio-Rad, Hercules, CA, USA).

### 4.5. Western Blotting

Total protein was extracted from plasma, placental tissues, and cells, which were lysed in RIPA buffer (WB3100, NCM Biotech, Suzhou, China) containing 1% phosphatase inhibitor (P003, NCM Biotech) and 1% proteinase inhibitor cocktail (HY-K0022, MedChemExpress). The protein concentration was determined using the BCA method (WB6501, NCM Biotech). The extracted protein lysate was denatured by adding 5 × SDS-PAGE loading buffer (WB2001, NCM Biotech) and heating at 100 °C for 10 min.

Equal amounts of protein were resolved by SDS-PAGE and transferred onto polyvinylidene fluoride (PVDF) membranes. Membranes were probed with the following primary antibodies: anti-APOA1 (dilution 1:5000, ab227455, Abcam, Cambridge, UK), anti-beta II Tubulin (dilution 1:10,000, ab151318, Abcam), and anti-albumin (dilution 1:2000, ab207327, Abcam) overnight at 4 °C. On the second day, the PVDF membrane was incubated with a horseradish peroxidase (HRP)-conjugated secondary antibody (dilution 1:10,000, 33101ES60, YEASEN) for 1 h at room temperature. Finally, the membrane was developed using the enhanced chemiluminescence method and visualized using an Amersham Imager 600 (GE Healthcare, Waukesha, WI, USA).

### 4.6. Immunohistochemistry (IHC)

Placental tissues were dissected and fixed in a 4% paraformaldehyde solution (P0099, Beyotime, Nantong, China). Tissues were then embedded in paraffin. Sections (5 μm) were cut and prepared for immunohistochemistry using standard procedures. Human placental sections were deparaffinized and treated with 3% H_2_O_2_ to block endogenous peroxidase activity. The sections were then pretreated with heat-mediated antigen retrieval using sodium citrate buffer or Tris/EDTA buffer (pH = 6.0) at 100 °C for 20 min, incubated with primary antibodies against APOA1 (dilution 1:500, ab227455, Abcam) at 4 °C overnight in a humidified chamber, and incubated with a secondary antibody (GB22301, Servicebio, Wuhan, China) for 30 min at room temperature. Reactions were visualized using DAB as a substrate (G1212, Servicebio). Images were acquired using a microscope (Olympus IX73, Olympus, Tokyo, Japan).

### 4.7. Luciferase Assay

The overexpressed *PPARγ* (PPAR*γ−OE*) and *APOA1* promoter plasmids were constructed by GeneChem (Shanghai, China). HEK293T or HTR8/SVneo cells were transfected with the overexpressed *PPARγ* plasmid and *APOA1* promoter plasmid using Lipofectamine 3000 (L3000075, Thermo Fisher Scientific) and cultured under standard culture conditions for 48 h according to the manufacturer’s protocol. The cells were then washed with PBS and lysed with cell lysis buffer (100 μL/well). The lysate was centrifuged at 10,000–15,000 rpm for 3–5 min, and the supernatant was transferred to a new tube for subsequent measurement. The relative light units (RLU) were measured by mixing 20 μL of sample supernatant and 20 μL Firefly Luciferase Assay Reagent using a dual-luciferase reporter assay system (GM-040503, Genomeditech, Shanghai, China). The RLU was measured again by mixing 20 μL of Renilla Luciferase Assay Reagent into the above measurement tube. The ratio of the two RLU measurements represents the activation of the reporter gene.

### 4.8. Statistical Analysis

All the experimental results were analyzed using GraphPad Prism 9.2.0 (GraphPad Software, San Diego, CA, USA). Friedman’s test was used to analyze variance. Differences between groups were analyzed using a one-way ANOVA and unpaired Student’s *t*-test. Results are presented as the mean ± standard deviation (SD), and the *p*-value < 0.05 was supposed to be statistically significant.

## Figures and Tables

**Figure 1 ijms-24-16363-f001:**
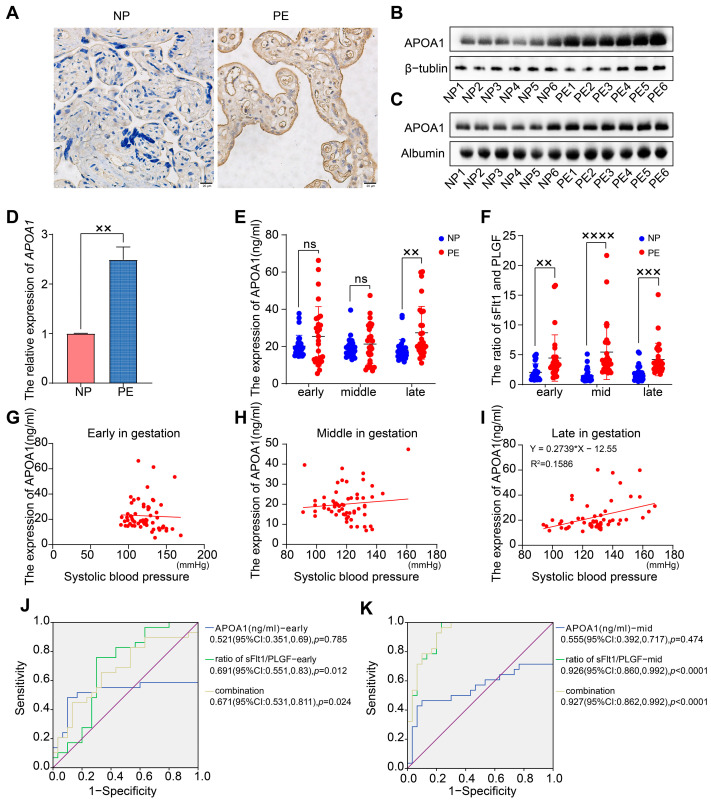
The expression of APOA1 is elevated in placental and plasma samples from women with preeclampsia. (**A**) Immunohistochemistry for APOA1 in placental tissues from NP (*n* = 12) and PE women (*n* = 12). (**B**) Western blot analysis of APOA1 protein levels in placental tissues from NP women (*n* = 6) and PE women (*n* = 6). (**C**) Western blot analysis of APOA1 protein levels in plasma tissues from NP (*n* = 6) and PE women (*n* = 6). (**D**) qRT-PCR analysis of *APOA1* in placental tissues from NP (*n* = 12) and PE women (*n* = 12). (**E**) ELISA assay for the expression of APOA1 in plasma tissues during early, mid-term, and late gestation from NP (*n* = 30) and PE (*n* = 29) patients. (**F**) ELISA assay for the ratio of sFlt1/PLGF in plasma tissues during early, mid-term, and late gestation from NP (*n* = 30) and PE (*n* = 29) patients. (**G**) Scatterplot of the level of APOA1 (ng/mL) in plasma tissues and systolic blood pressure during early gestation from 59 women. (**H**) Scatterplot of the expression level of APOA1 (ng/mL) in plasma tissues and systolic blood pressure during mid-term gestation from 59 women. (**I**) Scatterplot of the expression level of APOA1 (ng/mL) in plasma tissues and systolic blood pressure during late gestation from 59 women. (**J**) The ROC curve of the concentration of APOA1 and the ratio of sFlt1/PLGF during early gestation for predicting preeclampsia. (**K**) The ROC curve of the concentration of APOA1 and the ratio of sFlt1/PLGF during mid-term gestation for predicting preeclampsia. Error bars indicate mean ± SD. The data were analyzed using a one-way ANOVA assay and unpaired Student’s *t*-test. ^××^ *p*-value < 0.01, ^×××^ *p*-value < 0.001, ^××××^ *p*-value < 0.0001. ns, no significance.

**Figure 2 ijms-24-16363-f002:**
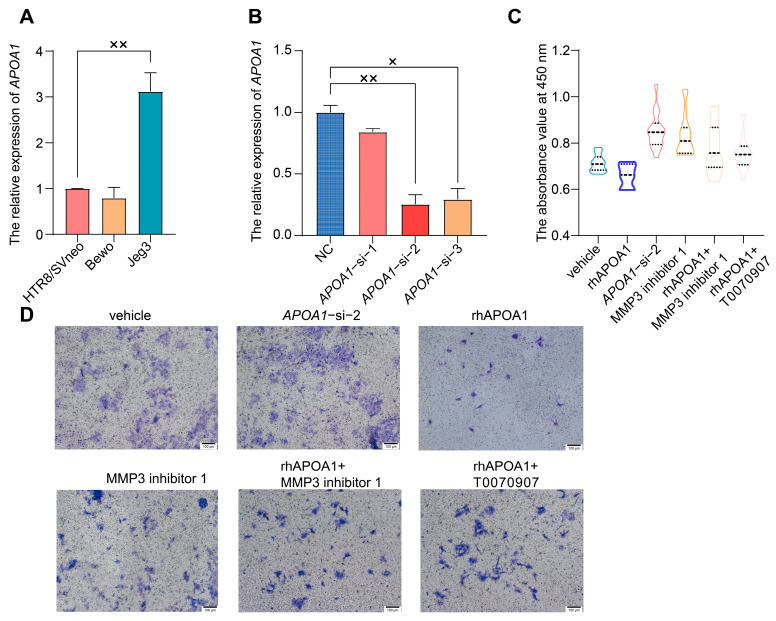
APOA1 can inhibit the proliferation and invasion of trophoblast cells. (**A**) qRT-PCR analysis of *APOA1* in Bewo, HTR8/SVneo, and Jeg3 cells. (**B**) qRT-PCR analysis of *APOA1* in HTR8/SVneo cells with negative control (NC) and *APOA1*-siRNA. (**C**) Proliferation function of the CCK-8 assay in vehicle, *APOA1*-si-2, rhAPOA1, MMP3 inhibitor 1, rhAPOA1 + MMP3 inhibitor 1, and rhAPOA1 + T0070907 groups. (**D**) Invasion ability of transwell assay in vehicle, *APOA1*-si-2, rhAPOA1, MMP3 inhibitor 1, rhAPOA1 + MMP3 inhibitor 1, and rhAPOA1 + T0070907 groups. Error bars indicate mean ± SD. The data were analyzed using a one-way ANOVA assay. ^×^ *p*-value < 0.05, ^××^ *p*-value < 0.01.

**Figure 3 ijms-24-16363-f003:**
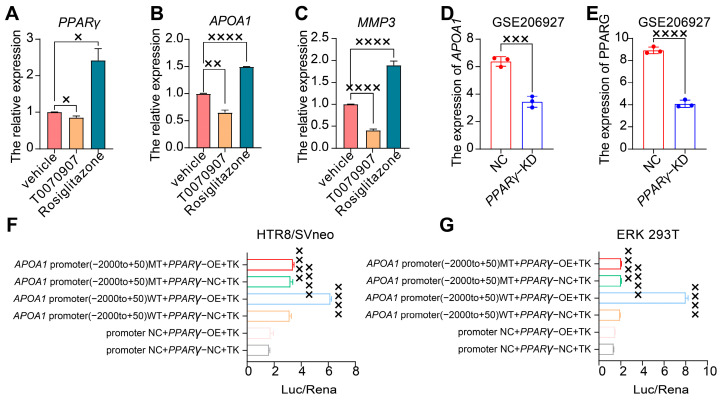
APOA1 was transcriptionally regulated by PPARγ. (**A**) qRT-PCR analysis of *PPARγ* in the vehicle, T0070907, and rosiglitazone groups. (**B**) qRT-PCR analysis of *APOA1* in vehicle, T0070907, and rosiglitazone groups. (**C**) qRT-PCR analysis of *MMP3* in vehicle, T0070907, and rosiglitazone groups. (**D**) The expression of *APOA1* in NC and *PPARγ*-KD cells from GSE206927 datasets. (**E**) The expression of *PPARγ* in NC and *PPARγ*-KD cells from GSE206927 datasets. (**F**) Dual luciferase assay for the transcriptional activity of PPARγ for the promoter of *APOA1* in HTR8/SVneo cells. (**G**) Dual luciferase assay for the transcriptional activity of PPARγ for the promoter of *APOA1* in HEK 293T cells. MT, mutation; WT, wild type; NC, normal control; OE, overexpressed. Error bars indicate mean ± SD. The data were analyzed using a one-way ANOVA assay and unpaired Student’s *t*-test. ^×^ *p*-value < 0.05, ^××^ *p*-value < 0.01, ^×××^ *p*-value < 0.001, ^××××^ *p*-value < 0.0001.

**Figure 4 ijms-24-16363-f004:**
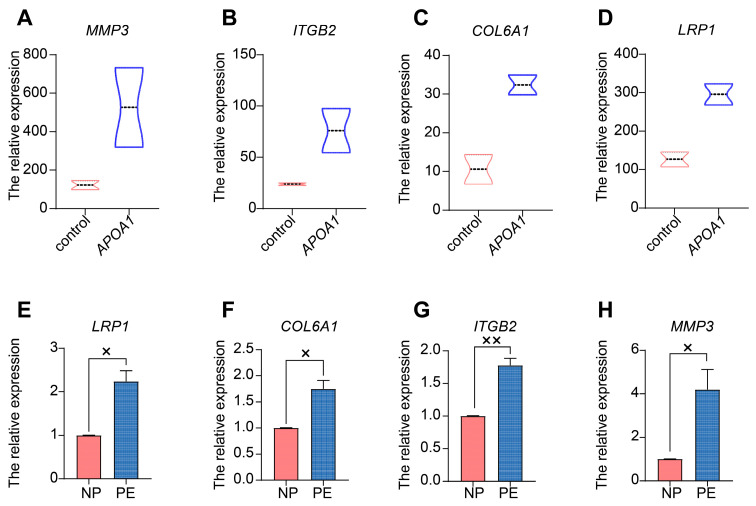
APOA1 functions by regulating several downstream targets. (**A**–**D**) Violin plots of *MMP3* (**A**), *ITGB2* (**B**), *COL6A1* (**C**), and *LRP1* (**D**) in control and *APOA1* overexpressed 4T cells from datasets. (**E**–**H**) qRT-PCR analysis of *LRP1* (**E**), *COL6A1* (**F**), *ITGB2* (**G**), and *MMP3* (**H**) in placental tissues from PE (*n* = 12) and NP (*n* = 12) women. Error bars indicate mean ± SD. The data were analyzed using an unpaired Student’s *t*-test. ^×^ *p*-value < 0.05, ^××^ *p*-value < 0.01.

**Figure 5 ijms-24-16363-f005:**
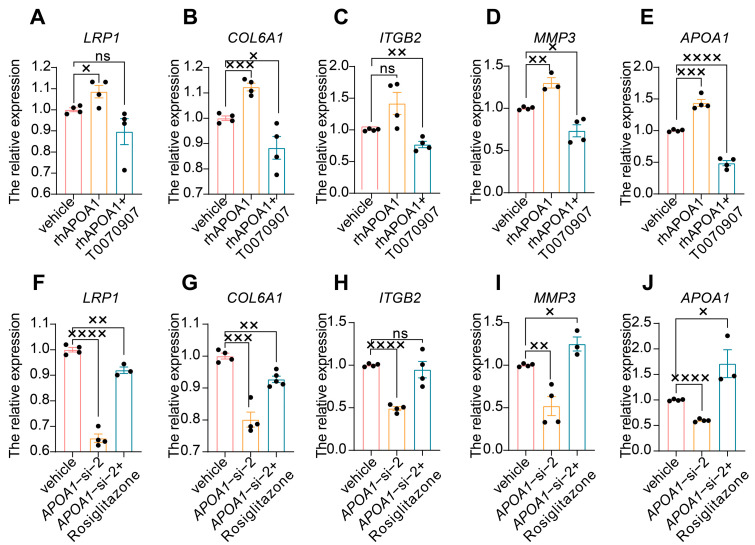
APOA1 functions by regulating several downstream targets. (**A**–**E**) qRT-PCR analysis of *LRP1* (**A**), *COL6A1* (**B**), *ITGB2* (**C**), *MMP3* (**D**), and *APOA1* (**E**) in vehicle, rhAPOA1, and rhAPOA1 + T0070907 groups. (**F**–**J**) qRT-PCR analysis of *LRP1* (**F**), *COL6A1* (**G**), *ITGB2* (**H**), *MMP3* (**I**), *APOA1* (**J**) in vehicle, *APOA1*-si-2, and *APOA1*-si-2 + rosiglitazone groups. Error bars indicate mean ± SD. The data were analyzed using a one-way ANOVA assay. ^×^ *p*-value < 0.05, ^××^ *p*-value < 0.01, ^×××^ *p*-value < 0.001, ^××××^ *p*-value < 0.0001. ns, no significance.

**Table 1 ijms-24-16363-t001:** Characteristics of the study population.

Characteristic	Normal Pregnancy (*n* = 30)	Preeclampsia (*n* = 30)	*p*-Value
Maternal age (years)	36.55 ± 2.587	37.07 ± 2.433	0.3435
Gestational age (weeks)	38.69 ± 1.161	36.25 ± 3.152	<0.0001 ****
Pregnancy BMI (kg/m^2^)	21.66 ± 2.880	24.53 ± 3.920	0.0003 ***
Systolic BP (mmHg)	110.1 ± 10.46	130.8 ± 12.91	<0.0001 ****
Diastolic BP (mmHg)	63.38 ± 8.999	81.12 ± 10.83	<0.0001 ****
Proteinuria (g/24 h)	-	0.8990 ± 1.433	-
Neonatal weight (g)	3500 ± 461.3	2785 ± 703.5	<0.0001 ****

All data are expressed as mean ± SD. *p*-values were obtained using the unpaired Student’s *t*-test in GraphPad Prism 9.2.0. *** *p*-value < 0.001, **** *p*-value < 0.0001. BP, blood pressure; BMI, body mass index.

**Table 2 ijms-24-16363-t002:** Primer sequences (all 5′-3′) used in qRT-PCR.

Gene	Primers	Sequence (5′-3′)
*GAPDH*	FORWARD	TTCGACAGTCAGCCGCATCTT
	REVERSE	CCCAATACGACCAAATCCGTT
*APOA1*	FORWARD	CTAAAGCTCCTTGACAACTGGG
	REVERSE	TTTCCAGGTTATCCCAGAACTC
*PPARγ*	FORWARD	AGATCATTTACACAATGCTGGC
	REVERSE	TAAAGTCACCAAAAGGCTTTCG
*LRP1*	FORWARD	AGTCTGCTTCGTGTGCCTATCC
	REVERSE	AGTCATTGTCATTGTCGCATCTCC
*COL6A1*	FORWARD	AGCACCTGGGCGTCAAAGTC
	REVERSE	TGTGGTCCGTGGCGATGATG
*ITGB2*	FORWARD	GGAGCAGCAGGACGGGATG
	REVERSE	GACGATGGCGGCGATGTTG
*MMP3*	FORWARD	CTTTCCTGGCATCCCGAAGTG
	REVERSE	CTCAACAGCAGAATCAACAGCATC

## Data Availability

Publicly available datasets were analyzed in this study. This data can be found here: [https://www.ncbi.nlm.nih.gov/geo/query/acc.cgi?acc=GSE202427 (accessed on 7 May 2023)], and [https://www.ncbi.nlm.nih.gov/geo/query/acc.cgi?acc=GSE206927 (accessed on 7 May 2023)].

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
