# Peer review of "APOA1 Is a Novel Marker for Preeclampsia"

_ijms, 2023, doi:10.3390/ijms242216363_

Round 1

Reviewer 1 Report

Comments and Suggestions for Authors

Article Review: "APOA1 is a novel marker for preeclampsia"

Summary: This basic science study delves into the investigation of APOA1 expression in the plasma and placental tissue of preeclamptic patients in comparison to non-preeclamptic individuals. The findings reveal an increased expression of APOA1 in preeclamptic patients. Notably, in vitro experiments demonstrated that an increase in APOA1 expression correlated with enhanced proliferation and invasion of trophoblastic cells.

Comments/Revisions:

1.       The study is methogolically sound with high quality data and is worthy of publication.

2.       The data sheds light on the pathogenesis of preeclampsia and carries potential clinical implications. It would be beneficial for the authors to elaborate on potential implications concerning the screening and monitoring of pregnant women, as well as the treatment of patients diagnosed with preeclampsia. A paragraph in the 'Discussion' section regarding the possible use of APOA1 levels for screening preeclampsia and the therapeutic targeting options for preeclamptic women is recommended. Additionally, the authors should clarify their proposed approach for targeting APOA1 in preeclamptic patients.

3.       In line 79 the word ‘perspective’ should be replaced with ‘prospective’

Recommendation: Minor revision

Comments on the Quality of English Language

Article Review: "APOA1 is a novel marker for preeclampsia"

Summary: This basic science study delves into the investigation of APOA1 expression in the plasma and placental tissue of preeclamptic patients in comparison to non-preeclamptic individuals. The findings reveal an increased expression of APOA1 in preeclamptic patients. Notably, in vitro experiments demonstrated that an increase in APOA1 expression correlated with enhanced proliferation and invasion of trophoblastic cells.

Comments/Revisions:

1.       The study is methogolically sound with high quality data and is worthy of publication.

2.       The data sheds light on the pathogenesis of preeclampsia and carries potential clinical implications. It would be beneficial for the authors to elaborate on potential implications concerning the screening and monitoring of pregnant women, as well as the treatment of patients diagnosed with preeclampsia. A paragraph in the 'Discussion' section regarding the possible use of APOA1 levels for screening preeclampsia and the therapeutic targeting options for preeclamptic women is recommended. Additionally, the authors should clarify their proposed approach for targeting APOA1 in preeclamptic patients.

3.       In line 79 the word ‘perspective’ should be replaced with ‘prospective’

Recommendation: Minor revision

Author Response

Response to Reviewer 1 Comments

Summary: This basic science study delves into the investigation of APOA1 expression in the plasma and placental tissue of preeclamptic patients in comparison to non-preeclamptic individuals. The findings reveal an increased expression of APOA1 in preeclamptic patients. Notably, in vitro experiments demonstrated that an increase in APOA1 expression correlated with enhanced proliferation and invasion of trophoblastic cells.

Point 1: 1. The study is methogolically sound with high quality data and is worthy of publication.

Response 1: Thank you most sincerely.

Point 2: 2. The data sheds light on the pathogenesis of preeclampsia and carries potential clinical implications. It would be beneficial for the authors to elaborate on potential implications concerning the screening and monitoring of pregnant women, as well as the treatment of patients diagnosed with preeclampsia. A paragraph in the 'Discussion' section regarding the possible use of APOA1 levels for screening preeclampsia and the therapeutic targeting options for preeclamptic women is recommended. Additionally, the authors should clarify their proposed approach for targeting APOA1 in preeclamptic patients.

Response 2: We gratefully appreciate for your valuable suggestion. We added a paragraph in “Discussion” section to discuss the possible use of APOA1 levels for screening and monitoring of pregnant women, as well as the treatment of patients diagnosed with preeclampsia. Additionally, we clarify the proposed approach for targeting APOA1 in preeclamptic patients. Please see the revised manuscript for details.

Point 3: 3. In line 79 the word ‘perspective’ should be replaced with ‘prospective’.

Response 3: Thank you for your correction, we have fixed the error.

Reviewer 2 Report

Comments and Suggestions for Authors

The idea of the study as well the topic are really important and up-to-date. It seems that nowadays we are seeking for early, tailored theranostics for our pregnant patients. The concept is clear as well proposed laboratory methods are obvious and often performed techniques. They are described in sufficient manner. 

However I have some doubts concerning methodology which have to solved/explained prior final decision... 

1. What was the time frame for early/middle/late pregnancy?

2. Which exactly definition of preeclampsia was used, is it surely the latest one? 

3. Did you combined patients with early and late PE? If yes that's not good, should be checked separately, but than we will decrease study group size, if presented patients are just from early or late PE, which do we have? 

4. What about analysis and differentiation between severity of PE? Did the patients had any pharmacotherapy, was it taken into account? 

5. Did any placental/sonographical parameters were analyzed? What about sflt/plgf? 

6. There is no explanation why other, and exactly those parameters/markers were analyzed in regard to APOA1? It should be clarified in the introduction part. 

7. For sure references are insufficiently developed. There are important papers which are not stated...

8. Some technical issues, e.g. In table 1. there is no-30, while in the text you stated 29. To just verified finally prior decision about publication. 

At this point I don't have any strong "cons" concerning techniques, but in such form; without very important data about study group we cannot formulate any convincing conslusions. To be clarified. 

Author Response

Response to Reviewer 2 Comments

The idea of the study as well the topic are really important and up-to-date. It seems that nowadays we are seeking for early, tailored theranostics for our pregnant patients. The concept is clear as well proposed laboratory methods are obvious and often performed techniques. They are described in sufficient manner.

However I have some doubts concerning methodology which have to solved/explained prior final decision...

Point 1: 1. What was the time frame for early/middle/late pregnancy?

Response 1: We gratefully appreciate for your valuable suggestion. In our study, we categorized the entire pregnancy into three stages, early pregnancy: up to (but not including) 14 weeks, mid-term pregnancy: between 14-27+6 weeks, and late pregnancy: 28 weeks and beyond. Furthermore, we also added the time frame in the “Methods” section, please see the revised manuscript for details.

Point 2: 2. Which exactly definition of preeclampsia was used, is it surely the latest one?

Response 2: We gratefully appreciate for your valuable suggestion. We used the latest definitions and diagnostic criteria for preeclampsia (with or without severe features) according to the American College of Obstetricians and Gynecologists (ACOG) (ref: 32443079).

Point 3: 3. Did you combined patients with early and late PE? If yes that's not good, should be checked separately, but than we will decrease study group size, if presented patients are just from early or late PE, which do we have?

Response 3: We gratefully appreciate for your valuable suggestion. Preeclampsia is a severe placenta-related pregnancy disorder that is generally divided into two subtypes named early-onset preeclampsia (onset <34 weeks of gestation), and late-onset preeclampsia (onset ≥34 weeks of gestation) (ref: 34992598). In our study, the women with preeclampsia were all diagnosis before 34 weeks, who were all diagnosed as early-onset preeclampsia.

Point 4: 4. What about analysis and differentiation between severity of PE? Did the patients had any pharmacotherapy, was it taken into account?

Response 4: We gratefully appreciate for your valuable suggestion. We recognize severe preeclampsia based on blood pressure (systolic blood pressure > 160 mmHg, and or diastolic blood pressure > 90 mmHg), or the presence of proteinuria. Diagnosed patients are treated with aspirin or Labetalol.

Point 5: 5. Did any placental/sonographical parameters were analyzed? What about sflt/plgf?

Response 5: We gratefully appreciate for your valuable suggestion. We didn’t analyze any placental/sonographical parameter. Additionally, we tested the expression level of sFLT and PLGF, and the ratio of sFLT/PLGF, please see the revised manuscript for details.

Point 6: 6. There is no explanation why other, and exactly those parameters/markers were analyzed in regard to APOA1? It should be clarified in the introduction part.

Response 6: We gratefully appreciate for your valuable suggestion. We explained why other, and exactly those parameters/markers were analyzed in regard to APOA1 in “Introduction” section.

Point 7: 7. For sure references are insufficiently developed. There are important papers which are not stated...

Response 7: We gratefully appreciate for your valuable suggestion. We checked the full manuscript and cited the important references. Please see the revised manuscript for details.

Point 8: 8. Some technical issues, e.g. In table 1. there is no-30, while in the text you stated 29. To just verified finally prior decision about publication.

Response 8: We gratefully appreciate for your valuable suggestion. In this study, we used data from 30 normal control women and 29 women with preeclampsia for analyzing. Additionally, we carefully checked the full manuscript, and corrected the error presentation.

At this point I don't have any strong "cons" concerning techniques, but in such form; without very important data about study group we cannot formulate any convincing conslusions. To be clarified.

Reviewer 3 Report

Comments and Suggestions for Authors

The paper by Zhenzhen et al. focuses on the role of APOA1 in preeclampsia using a variety of experimental approaches. The topic is of great interest and the paper is overall well written. However, results of most experiments are of poor quality (e.g. IHC, western blot on "plasma tissue") and conclusions are too strong compared to the evidence of the obtained results. I would be more cautious on results obtained and I would not generalise the results to the complex clinical scenario of preeclampsia. I recommend Authors to perform other experiments and do another ex novo submission of the paper since in its current form is not acceptable for publication.

Author Response

Response to Reviewer 3 Comments

Point 1: The paper by Zhenzhen et al. focuses on the role of APOA1 in preeclampsia using a variety of experimental approaches. The topic is of great interest and the paper is overall well written. However, results of most experiments are of poor quality (e.g. IHC, western blot on "plasma tissue") and conclusions are too strong compared to the evidence of the obtained results. I would be more cautious on results obtained and I would not generalise the results to the complex clinical scenario of preeclampsia. I recommend Authors to perform other experiments and do another ex novo submission of the paper since in its current form is not acceptable for publication.

Response 1: We gratefully appreciate for your valuable suggestion. We have performed other experiments and make our results and conclusions more perfect.

Round 2

Reviewer 2 Report

Comments and Suggestions for Authors

I see sufficient improvement in the revised paper.

All my concerns were addressed.

In the present form manuscript can be published.

Author Response

Point 1: I see sufficient improvement in the revised paper.

              All my concerns were addressed.

              In the present form manuscript can be published.

Response 1: Thank you most sincerely.

Reviewer 3 Report

Comments and Suggestions for Authors

I commend the Authors for their nice revision. However, I still have major concerns.

When I asked Authors more experiments I meant replications of the already performed experiments (e.g. more images of IHC for APOA1 in placental tissue, other WB) since the quality of images is quite low.

Authors added ELISA assay for APOA1, which is significantly more accurate than WB, but they did not mention it in Results Section 2.1. 

Figure 4 A B C and D are not necessary and may be included as supplementary material. 

Figure 5 needs an extensive explanation both in figure description and results section. 

Author Response

Response to Reviewer 3 Comments

I commend the Authors for their nice revision. However, I still have major concerns.

Point 1: When I asked Authors more experiments I meant replications of the already performed experiments (e.g. more images of IHC for APOA1 in placental tissue, other WB) since the quality of images is quite low.

Response 1: We gratefully appreciate for your valuable suggestion. We performed more replicate experiments, including IHC and WB, for APOA1. Please refer to the revised manuscript and Figure 1 for further details.

Point 2: Authors added ELISA assay for APOA1, which is significantly more accurate than WB, but they did not mention it in Results Section 2.1.

Response 2: Thanks for your suggestions. We have discussed the results of ELISA assay for APOA1 in Results Section 2.1. Please see the revised manuscript for details.

Point 3: Figure 4 A B C and D are not necessary and may be included as supplementary material.

Response 3: We gratefully appreciate for your valuable suggestion. We have moved Figure 4 A B C, and D to the supplementary material.

Point 4: Figure 5 needs an extensive explanation both in figure description and results section.

Response 4: We appreciate your valuable suggestion. We have added an extensive explanation in both the figure description and results sections in Figure 5. Please refer to the revised manuscript for further details.

Round 3

Reviewer 3 Report

Comments and Suggestions for Authors

Authors made the appropriate revisions and I now believe the paper is suitable for publication